# POD1-SUN-CRT3 chaperone complex guards the ER sorting of LRR receptor kinases in Arabidopsis

Yong Xue[1,3], Jiang-Guo Meng[1,2,3], Peng-Fei Jia[1], Zheng-Rong Zhang[1,2], Hong-Ju Li [1,2✉] & Wei-Cai Yang [1,2✉]

Protein sorting in the secretory pathway is essential for cellular compartmentalization and homeostasis in eukaryotic cells. The endoplasmic reticulum (ER) is the biosynthetic and folding factory of secretory cargo proteins. The cargo transport from the ER to the Golgi is highly selective, but the molecular mechanism for the sorting specificity is unclear. Here, we report that three ER membrane localized proteins, SUN3, SUN4 and SUN5, regulate ER sorting of leucine-rich repeat receptor kinases (LRR-RKs) to the plasma membrane. The triple mutant *sun3/4/5* displays mis-sorting of these cargo proteins to acidic compartments and therefore impairs the growth of pollen tubes and the whole plant. Furthermore, the extra-cellular LRR domain of LRR-RKs is responsible for the correct sorting. Together, this study reports a mechanism that is important for the sorting of cell surface receptors.

[1] State Key Laboratory of Molecular Developmental Biology, Institute of Genetics and Developmental Biology, Chinese Academy of Sciences, Beijing 100101, China. [2] University of Chinese Academy of Sciences, Beijing 100049, China. [3] These authors contributed equally: Yong Xue, Jiang-Guo Meng. ✉email: hjli@genetics.ac.cn; wcyang@genetics.ac.cn

 **1**

In eukaryotic cells, the ER is the entry organelle of membrane and secretory proteins. In the ER, folding, sorting and trafficking of cargo proteins to the right destination is critical for the buildup of cells. The newly synthesized cargoes are inserted into the ER as nascent polypeptides through the translocation machinery. Right after translocation, protein folding starts through the assistance of molecular chaperones, disulfide bond formation and N-gylcosylation events, to prevent protein aggregation and promote proper folding into the native state[1]. Two major protein-folding systems, the BiP and calnexin/calreticuliin (CNX/CRT) pathways, execute the chaperone-assisted formation of the tertiary conformation of cargoes[2]. BiPs are ER-localized heat shock protein 70 (HSP70) molecular chaperones. CNXs and CRTs are homologous lectins that interact with newly synthesized glycoproteins in the ER and serve as molecular chaperones to assist protein folding and retain folding intermediates to prevent secretion[1]. Two CNXs (CNX1 and CNX2) and three CRTs (CRT1, CRT2, and CRT3) are encoded by Arabidopsis genome[3]. The N-glycosylated cargoes enter into the CNX/CRT folding cycle for folding/refolding and the following exit for sorting or degradation through recognition of the glycosylation moieties by chaperones, a process called ER quality control (ER-QC)[4,5]. The finally misfolded/unfolded proteins would be degraded by the ERAD pathway, and the properly folded/assembled cargoes are packaged into the COPII vesicles and sorted to the Golgi apparatus for further sorting to the plasma membrane or vacuole/lysosome pathway[6–9]. Two ER exit mechanisms, namely, bulk flow and cargo receptor capture, are used by different cargoes[10]. The binding of SEC24A to the cytoplasm-facing export motifs of the cargoes initiates the sequential assembly of COPII machinery[11]. The topology of the membrane cargoes has been determined during entry into the ER, but the cytosolic COPII components assemble and initiate vesicle budding only after proper cargo folding in the ER lumen. An obvious missing link here, which is not well understood, is how different cargoes and their folding state are discriminated for export.

POLLEN DEFECTIVE IN GUIDANCE1 (POD1) is a conserved ER-localized protein with unknown biochemical activity and was previously identified to be required for navigating the pollen tubes to the ovules in Arabidopsis[12]. Based on the interaction of POD1 with the ER chaperone CRT3, which is required for protein maturation in plants[13,14], POD1-CRT3 complex has been postulated to regulate the folding and maturation of cargo proteins in pollen tubes[2,12]. However, the molecular function and mechanism of POD1-CRT3 machinery in the ER is still unclear. Here we report three ER membrane proteins SUN3, SUN4, and SUN5 in a complex with POD1 to be required for the ER sorting of LRR receptors to the plasma membrane, which depends on the extracellular LRR domains.

## Results

### SUN proteins evolutionarily diverged into two clusters in eukaryotes.
POD1 was previously identified to be localized in the ER and modulate the male response to female signals during pollen tube guidance in Arabidopsis[12]. Its yeast orthologue EMP65 was identified to be in an ER protein complex with SUN-like protein (Slp1)[12,15–17]. The Slp1 ortholog in mice was also reported to be localized in the ER[18]. Phylogenetic analysis showed that the SUN domain-containing proteins in eukaryotes are grouped into two clusters. Cluster I harbor a C-terminal SUN domain and variable numbers of N-terminal transmembrane domains (TMs), whereas cluster II contains a middle SUN domain spanned by TMs at both the N- and C-terminal ends (Fig. 1a, Supplementary Fig. 1). The Arabidopsis genome encodes five SUN-domain proteins, among which AtSUN1 and AtSUN2

belong to Cluster I and group with the yeast Msp3 and human SUN proteins, while AtSUN3, AtSUN4, and AtSUN5 belong to Cluster II and group with the yeast Slp1 and mouse Osteopotentia (OSPT). The SUN domain of AtSUN1 and AtSUN2 resides in the C-terminal preceded by a coiled-coil (CC) domain, with only one TM at the N-terminal end, whereas AtSUN3, AtSUN4, and AtSUN5, have a SUN domain located in the middle followed by a CC domain, and one N-terminal and two C-terminal TMs (Fig. 1a, b, Supplementary Fig. 1). AtSUN1 and AtSUN2 are specifically localized on the nuclear envelope (NE), and the nuclei in sun1 sun2 mutant root cells are rounder than the wild-type (WT), although the physiological significance of nuclear shape is still unclear[19–21]. In contrast, ectopically expressed AtSUN3 and AtSUN4 were shown to be localized on both the NE and ER in tobacco leaves[20]. The distinct topology, domain architecture, phylogenetic grouping, and subcellular localization of the two clusters implies an early functional divergence during evolution.

To elucidate the possible association between POD1 and SUN3, SUN4, and SUN5 in plants, co-immunoprecipitation (Co-IP) assay with Arabidopsis leaf protoplasts was performed. The results show that the three SUN proteins are physically associated with POD1, but not with CNX1, one of the five CNX/CRT chaperones in Arabidopsis (Fig. 1c–e). This result indicates that POD1 and SUN3/4/5 are in the same complex through direct or indirect association. Subcellular fractionation experiments showed that both SUN4-GFP and CNX1 are enriched in the membrane fraction ($P_{100}$) (Fig. 1f). N-linked protein glycosylation in the ER occurs to many ER cargoes and resident proteins. PNGase F is an endoglycosidase that cleaves the N-linked glycans from glycoprotein[22]. PNGase F treatment showed that tagged SUN3 and SUN4 proteins expressed in Arabidopsis protoplasts can be cleaved and exhibit a shift in the gel blot assay (Fig. 1g). This suggests that SUN4 is associated with ER membrane and most likely glycosylated.

To study the biological function of SUN3, SUN4, and SUN5, we examined their expression in pollen. RT-PCR results showed that SUN3, SUN4, and SUN5 are expressed in inflorescences, mature flowers, pollen grains right before anthesis (stage 12c) and mature pollen (Fig. 1h). A previous study failed to generate viable triple mutant due to embryogenesis defect at the condition that single and double mutants exhibit no obvious phenotype[20]. However, luckily, we obtained two independent viable alleles of the T-DNA insertion triple mutants of sun3 sun4-1 sun5 and sun3 sun4-2 sun5 in Col-0 ecotype background by crossing. The T-DNAs are all inserted in the exons and generated null mutations as shown by RT-PCR (Fig. 1i, j). This provides an opportunity for the functional study of these genes. No obvious morphological defect was observed under normal growth conditions for sun3, sun4-1, sun4-2, sun5 single mutants, and sun3 sun4-1, sun3 sun4-2, sun3 sun5, sun4-1 sun5, and sun4-2 sun5 double mutants. Both sun3 sun4-1 sun5 and sun3 sun4-2 sun5 triple homozygous mutants exhibit the same growth and reproductive defects, as described below. The above genetic analysis indicates that these three genes function redundantly in regulating plant growth and reproduction. Hereafter, sun3/4/5 (short for sun3 sun4-1 sun5) was used as a representative for the further study.

### sun3/4/5 displays defective reproductive and vegetative growth.
In sun3/4/5, shortened stamen filaments make the plant sterile at natural condition (Fig. 2a, b, Supplementary Fig. 2). Thus, hand-pollination was necessary for offspring production and transgenic experiments. Even at hand-pollination condition, the siliques of sun3/4/5 are shorter than the WT (Fig. 2c). Reciprocal analysis between the (WT) and sun3/4/5 showed reduced seed set when

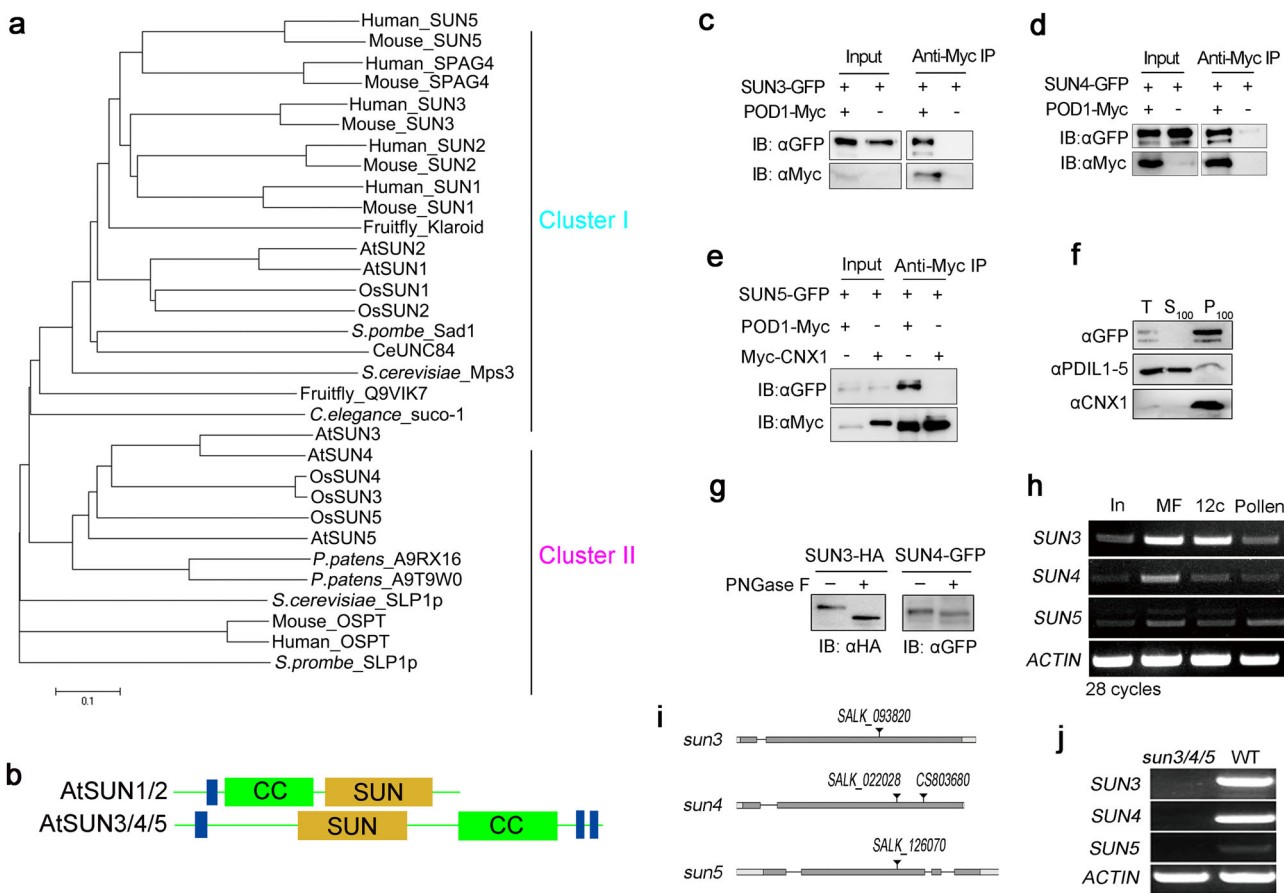

**Fig. 1 SUN proteins interact with POD1. a** Phylogenetic tree of SUN proteins. **b** Domain structure of AtSUN1/2 and AtSUN3/4/5. **c–e** SUN3, 4, and 5 interact with POD1 in Arabidopsis protoplasts by Co-IP, but not with ER chaperone CNX1. **f** SUN4-GFP is detected in the microsome fractions ($P_{100}$) after cellular fractionation . PDIL1-5 and CNX1 were used as the control of soluble ($S_{100}$) and membrane protein ($P_{100}$), respectively. T, total protein. **g** SUN3 and SUN4 are N-glycosylated proteins. Total proteins extracted from protoplasts expressing SUN3-HA and SUN4-GFP, respectively, were treated with (+) or without (-) N-glycosidases PNGase F and probed for the GFP tag. **h** Transcripts of *SUN3*, *SUN4*, and *SUN5* were detected in inflorescences, mature flowers, pollen at stage 12c and mature pollen by RT-PCR. **i** Schematic diagrams of the gene structures of *SUN3*, *SUN4*, and *SUN5*, and the T-DNA insertion sites of mutants. Dark grey, exons; light grey, UTRs; lines, introns; triangle, the T-DNA insertion site. **j** The transcripts of *SUN3/4/5* were not detected in *sun3/4/5* triple mutant by RT-PCR. 3 (**c–h**) and 5 (**j**) independent biological experiments were repeated.

*sun3/4/5* was used as the male plant, but only slightly as the female (Fig. 2d, e). In self-pollination and cross pollination using *sun3/4/5* as the male plant, seed set was significantly reduced mostly in the basal half, but not in the apical half (Fig. 2d, e). Male transmission efficiency of each mutation in the triple mutant background was significantly reduced (Supplementary Table 1). The in vitro and in vivo pollen tube germination, tube growth, and pollen tube targeting to the WT ovule was reduced in *sun3/4/5* compared with the WT (Fig. 2f and Supplementary Fig. 3). These results suggest that *sun3/4/5* impairs the male gametophyte function but not the female gametophyte.

To verify the male gametophytic defect of *sun3/4/5*, *SUN4-GFP* fusion construct was expressed in the pollen and pollen tube driven by the pollen-specific promoter *LAT52* in the mutant. The result showed that the fluorescent fusion protein was localized on tubular structures in pollen and pollen tubes (Fig. 2g–k). DAPI staining of nuclei revealed that SUN4-GFP exhibits no obvious nuclear envelope accumulation, in contrast to its paralogs AtSUN1 and AtSUN2 as previously reported[19] (Fig. 2g–j). This fusion construct driven by *LAT52* promoter can rescue the seed set defect of *sun3/4/5* (Fig. 2l). No GFP signal was observed in transgenic plants expressing SUN4-GFP driven by its native promoter, nevertheless, defects of fertility and vegetative growth were rescued (Fig. 2l). This implies that

the level of SUN proteins is under stringent control in plants. Further analysis showed that ectopically overexpressed SUN3-GFP, SUN4-GFP and SUN5-GFP driven by 35 S promoter were localized to the ER and nuclear envelope in tobacco leaf epidermal cells (Fig. 2m, Supplementary Fig. 4), which is consistent with the previously reported[20]. Thus, it cannot be excluded that SUN3/4/5 also function on the nuclear membrane, which is continuous with the peripheral ER. Furthermore, to investigate the function of the SUN and coiled-coil (CC) domains, these two domains were replaced with GFP, respectively, and expressed in tobacco. The results showed that deletion of the SUN domain impaired the translation or stability of the protein, as no fluorescence was observed in repeated experiments (Supplementary Fig. 5). Interestingly, deletion of the CC domain leads to mislocalization of the protein to the Golgi, implying that the CC domain is required for the ER retention (Supplementary Fig. 5). In summary, SUN3/4/5 are ER-resident proteins and required for pollen function.

Additionally, *sun3/4/5* also exhibits pleiotropic phenotypes, such as small seedlings, shortened roots and dwarf plant architecture, compared with the WT (Fig. 3a–c). *SUN4-GFP* driven by either the constitutive 35 S promoter or its native promoter completely rescued the vegetative growth defect (Fig. 3b, c). Measurement of

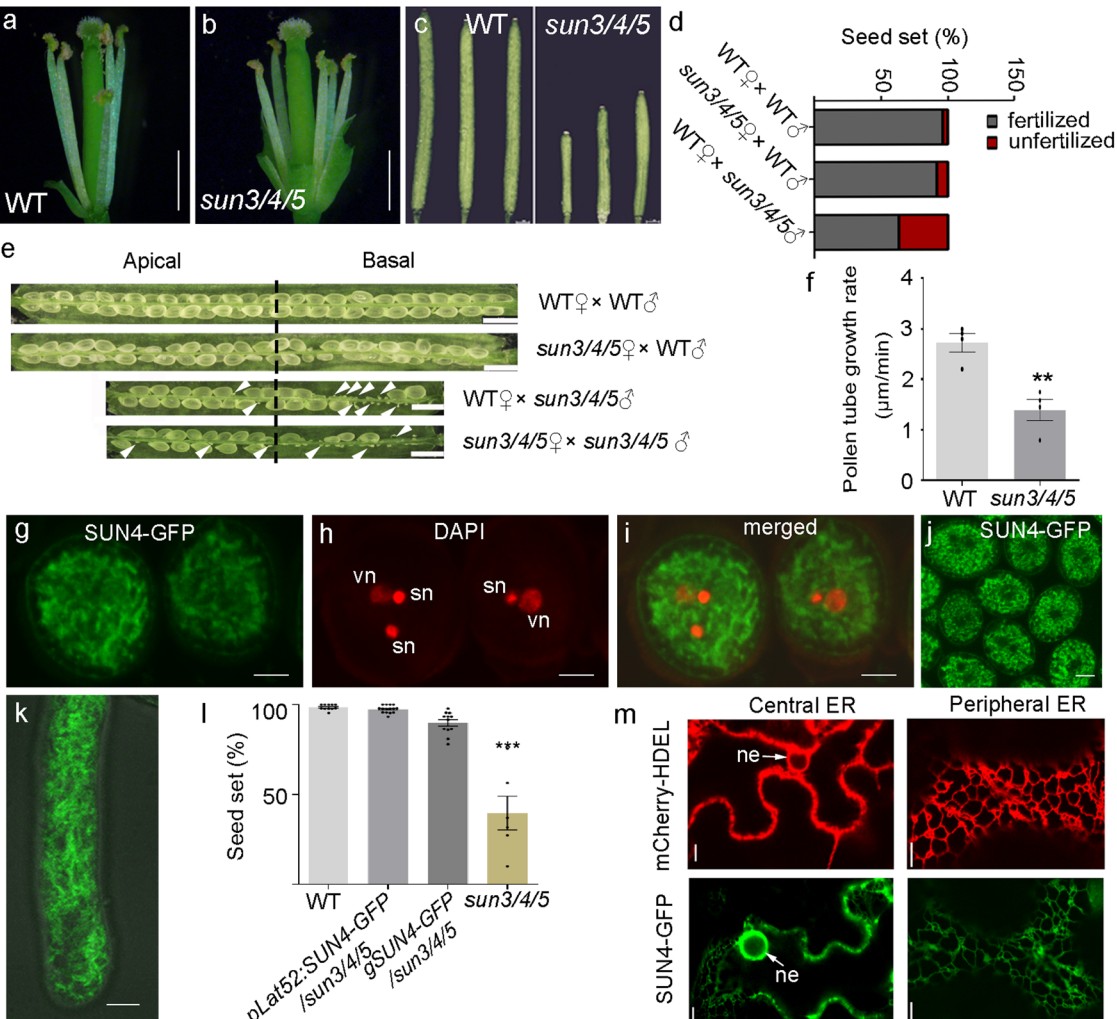

**Fig. 2 Reproductive phenotype of *sun3/4/5*. a** Stamen filaments in WT. **b** Stamen filaments in *sun3/4/5*. Bar, 1 mm. **c** The siliques are shortened in *sun3/4/5*. The siliques are imaged 14 days after hand pollination. Bar, 1 mm. **d**, **e** Seed set of selfed and crossed *sun3/4/5*. Arrowhead, aborted ovules. Bar, 1 mm. **f** Pollen tube growth rate. Bars represent means ± s.e.m. Two-tailed Student's *t*-test, $p^{**} = 0.0029$, $n = 4$ pollen tubes of WT and *sun3/4/5*, respectively, were measured. **g–k** Expression of SUN4-GFP driven by the *LAT52* promoter in pollen grains. 6 independent biological experiments were repeated. **g** GFP channel. **h** DAPI staining showing the two sperm nuclei (sn) and vegetative nucleus (vn). **i** The merged image of **g** and **h**. **j** SUN4-GFP in pollen grains. **k** SUN4-GFP in pollen tubes. Bar, 5 μm. **l** Genetic rescue of the *sun3/4/5* by *SUN4-GFP* driven by the native promoter and *LAT52* promoter, respectively. More than 30 siliques from 6 independent plants of each transgenic line were measured. The values represent means ± s.e.m., Two-tailed Student's *t*-test, $p^{***} < 0.001$. **m** Expression of SUN4-GFP in tobacco leaf epidermal cells shows ER pattern like that of mCherry-HDEL marker. ne, nuclear envelope. Bar, 10 μm. 6 independent biological experiments were repeated.

the root length after propidium iodide staining showed that both the division zone (DZ) and elongation zone (EZ) of *sun3/4/5* roots are shorter than those of the WT (Fig. 3d). The root hairs (RH) length, cell size and number in the division and elongation zone are also reduced (Fig. 3e–g). The root hairs appear extremely dense at the mature zone due to reduced root cell size (Fig. 3h). These data suggest that SUN3, SUN4, and SUN5 redundantly regulate vegetative growth by affecting cell division and expansion.

**SUN3, SUN4, and SUN5 regulate sorting of LRR-RKs.** It was speculated that POD1 likely regulates the folding or processing of cell surface receptors, based on its ER localization and association with ER chaperone CRT3 evidenced by yeast two-hybrid and BiFC assay[2,12]. Co-IP assay with Arabidopsis protoplasts further confirmed the association between POD1 and CRT3 (Fig. 4a). Co-IP result showed that SUN4 interacts with CRT3 specifically as well, but not with CRT2, and this interaction is not disrupted by tunicamycin (Tm), an inhibitor of the glycosylation process in

the ER (Fig. 4b). This suggests that their interaction does not require protein glycosylation. CRT3 contains three domains, the N-terminal lectin domain for glycan-binding, the middle P domain and the C-terminal tail (Fig. 4c). Co-IP with these three fragments showed that the lectin domain is responsible for the interaction with SUN4 (Fig. 4d). These results indicate a physical association between SUN-POD1 and CRT3.

In the ER, protein folding is tightly linked with ER exit process through which only properly folded and assembled proteins can be sorted to the COPII vesicles for delivery to the Golgi trafficking route. In Arabidopsis, the plant-specific CRT3 retains the misfolded cell-surface brassinosteroid (BR) receptor BRASSI-NOSTEROID INSENSITIVE 1 (BRI1) in the ER, and regulates the maturation of bacterial elongation factor Tu receptor EFR through direct binding[13,14,23–25]. Both BRI1 and EFR are leucine-rich repeat receptor kinases (LRR-RKs) with an extracellular ligand-perceiving LRR domain that undergoes chaperone-assisted protein folding in the ER lumen. It is tempting to speculate that

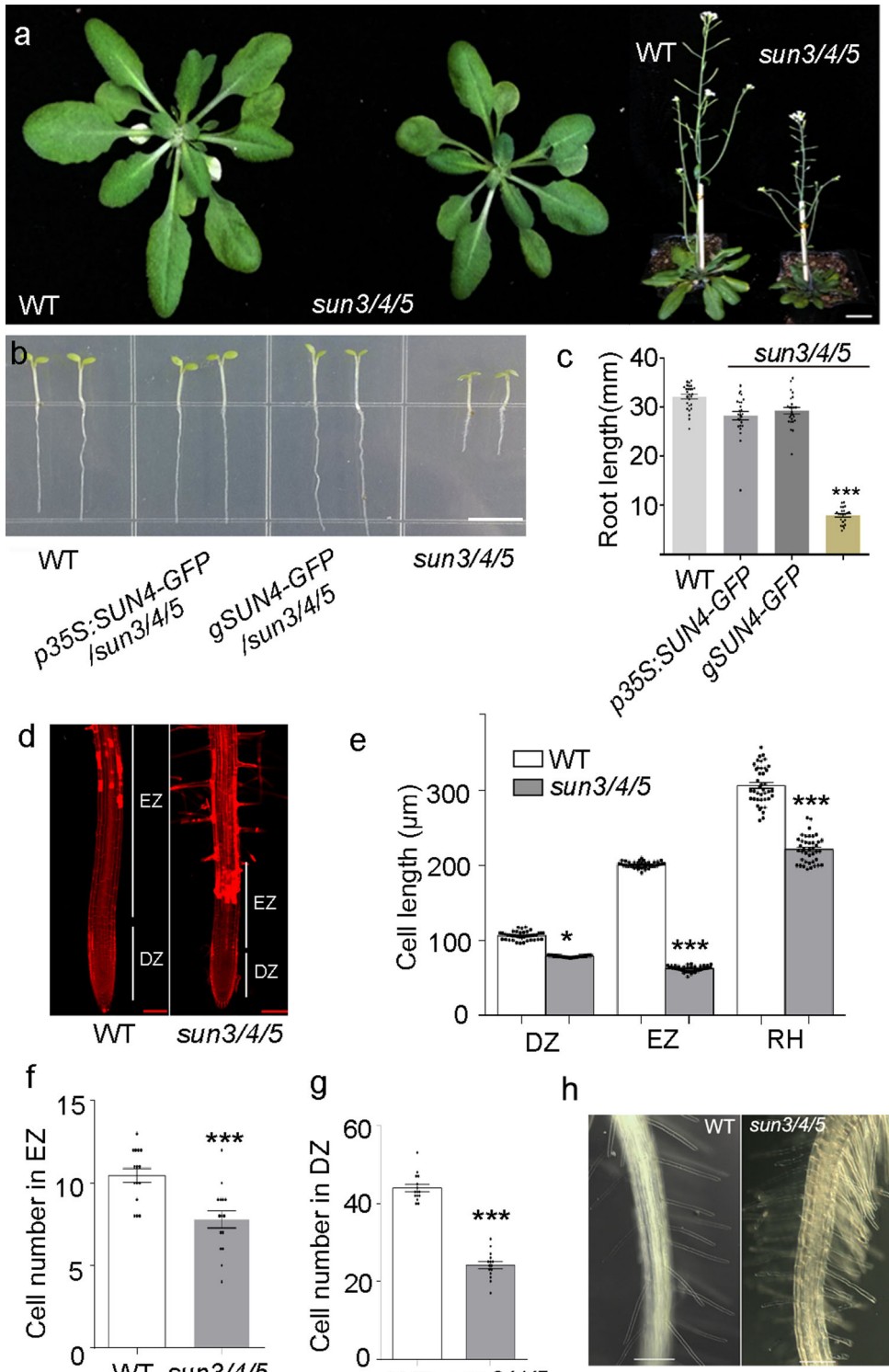

**Fig. 3 Vegetative phenotype of *sun3/4/5*. a** *sun3/4/5* plants are smaller and dwarf than the WT. Bar, 2 cm. **b** The shortened root of *sun3/4/5* can be rescued by *SUN4-GFP* driven by the *35 S* and *SUN4* promoters, respectively. The seedlings were grown for 7 days after germination (DAG). Scale bar, 0.5 cm. **c** Statistics of root length of different genotypes. More than 300 seedlings from 6 independent plants of each transgenic line were measured at 7 DAG. The values represent means ± s.e.m., Two-tailed Student's *t*-test, $p^{***} < 0.001$. **d** Division and elongation zone of WT and *sun3/4/5* roots at 7 DAG stained with propidium iodide. DZ, division zone; EZ, elongation zone. 3 independent biological experiments were independently repeated. Scale bar, 100 μm. **e** Cell length in different zones of WT and *sun3/4/5* roots at 7 DAG. RH, root hair. The values represent means ± s.e.m., Two-tailed Student's *t*-test, $p^* < 0.05$, $p^{***} < 0.001$. For DZ measurements, $n = 40$ and 37 separately. For EZ measurements, $n = 38$ and 40 separately. For RH measurements, $n = 41$ and 40 separately. **f** Statistics of cell numbers in the elongation zone. 15 seedlings of WT and *sun3/4/5* were measured at 7 DAG. The values represent means ± s.e.m., Two-tailed Student's *t*-test, $p^{***} < 0.001$. **g** Statistics in cell number in division zone. 15 seedlings of WT and *sun3/4/5* were measured at 7 DAG. The values represent means ± s.e.m., Two-tailed Student's *t*-test, $p^{***} < 0.001$. **h** Root hairs. 3 independent biological experiments were repeated. Bar, 100 μm.

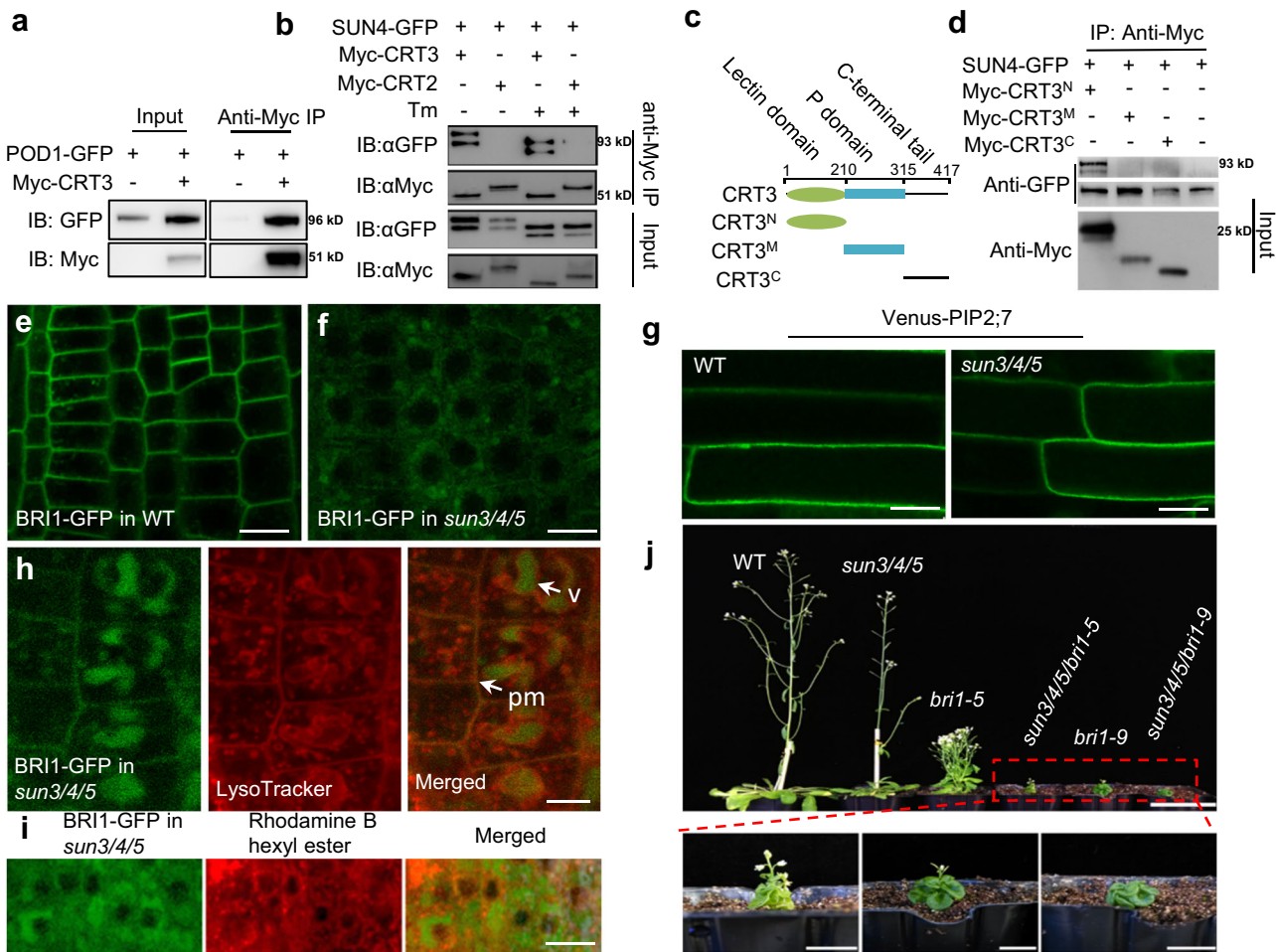

**Fig. 4 Plasma membrane targeting of BRI1 is impaired in *sun3/4/5*. a** POD1 interacts with CRT3 by Co-IP assay with Arabidopsis protoplasts. **b** SUN4 interacts with CRT3 independently of glycosylation. The protoplasts were treated with 5 mM Tm to inhibit the protein glycosylation after transformation with the indicated plasmids. The shifted bands indicated that the glycosylation of SUN4 was inhibited. CRT2 was used as a negative control. **c** The protein structure of CRT3 and the constructs used in **d**. **d** The interaction between SUN4 and CRT3 depends on the lectin domain. **e, f** Confocal image of BRI1-GFP in the WT (**e**) and *sun3/4/5* (**f**) root cells. Scale bar, 10 μm. **g** Confocal image of Venus-PIP2;7 in the WT and *sun3/4/5* root cells. Scale bar, 10 μm. **h** LysoTracker red staining of BRI1-GFP in *sun3/4/5*. Left panel, GFP channel; middle panel, LysoTracker red staining; right panel, merged image. Arrow, vacuole (v) and the plasma membrane (pm). Bar, 5 μm. **i** Rhodamine B hexyl ester staining of the ER in roots of *BRI1-GFP; sun3/4/5*. Left panel, GFP channel; middle panel, Rhodamine B hexyl ester staining; right panel, merged image. Bar, 5 μm. **j** Loss of *SUN3*, *SUN4* and *SUN5* aggravates the growth defect of *bri1-9* and *bri1-5*. 3 (**a**, **b**, **d**) and 6 (**e–i**) independent biological experiments were repeated.

SUN3/4/5 proteins also regulate the processing of similar cell surface receptors.

To test this hypothesis, we first examined the localization of BRI1-GFP, as a marker, in *sun3/4/5* and used a transmembrane aquaporin Venus-PIP2;7 as the control. Both proteins are synthesized in the ER and transported from the Golgi to the plasma membrane (PM)[26]. The result showed that the PM localization of BRI1-GFP in roots was largely abolished in *sun3/4/5*, but the localization of PIP2;7 showed no difference between the WT and mutant (Fig. 4e–g). In the WT root, BRI1-GFP labels the PM and endosomes, whereas in *sun3/4/5*, the GFP signal on the PM is very weak and some intracellular vesicles larger than endosomes are labeled by BRI1-GFP. A significant portion of these larger vesicles reside in acidic compartments stained by acidophilic dye LysoTracker Red[27], but not the ones stained by rhodamine B hexyl ester, which labels the ER in plant cells[28] (Fig. 4h, i). This indicates that in *sun3/4/5* cargo proteins are mis-sorted to acidic compartments, but not retained in the ER. This is distinct from the ER retention of receptors or other cargoes caused by the depletion of ER-QC components or mutations affecting cargo folding. It was reported that the folding-defective point-mutated bri1-5 and bri1-9 are retained in the ER by enhanced interaction with CRT3 and can be released to the PM and recover the plant defect in *crt3* mutant that is defective in ER-QC[14]. To examine the effect of SUNs on the processing of mutated BRI1s, we introduced *bri1-5* and *bri1-9* into the *sun3/4/5* background by crossing. In contrast to the effect of *crt3*, *sun3/4/5* aggravates the growth defect of *bri1-5* and *bri1-9*, respectively (Fig. 4j). This suggests that the export of mutated BRI1 to the plasma membrane is further blocked in *sun3/4/5*[14]. And this implies other pathways are also involved in the secretion of BRI1, which confers partial PM targeting that sustains the plant growth. Furthermore, BR-induced hypocotyl assay[29] showed that BR treatment cannot rescue the shortened hypocotyl of *sun3/4/5*, and *sun3/4/5* showed reduced sensitivity to high BR concentration that inhibits the hypocotyl elongation (Supplementary Fig. 6). This corroborates the phenotype of the reduced presence of BRI1 on the PM of *sun3/4/5*. RT-PCR results showed that *CRT3* expression remains unchanged in *sun3/4/5*, but other ER chaperones, *BiP1*, *BiP2*, *CRT1*, *CNX1*, and *PDIL1-5* are

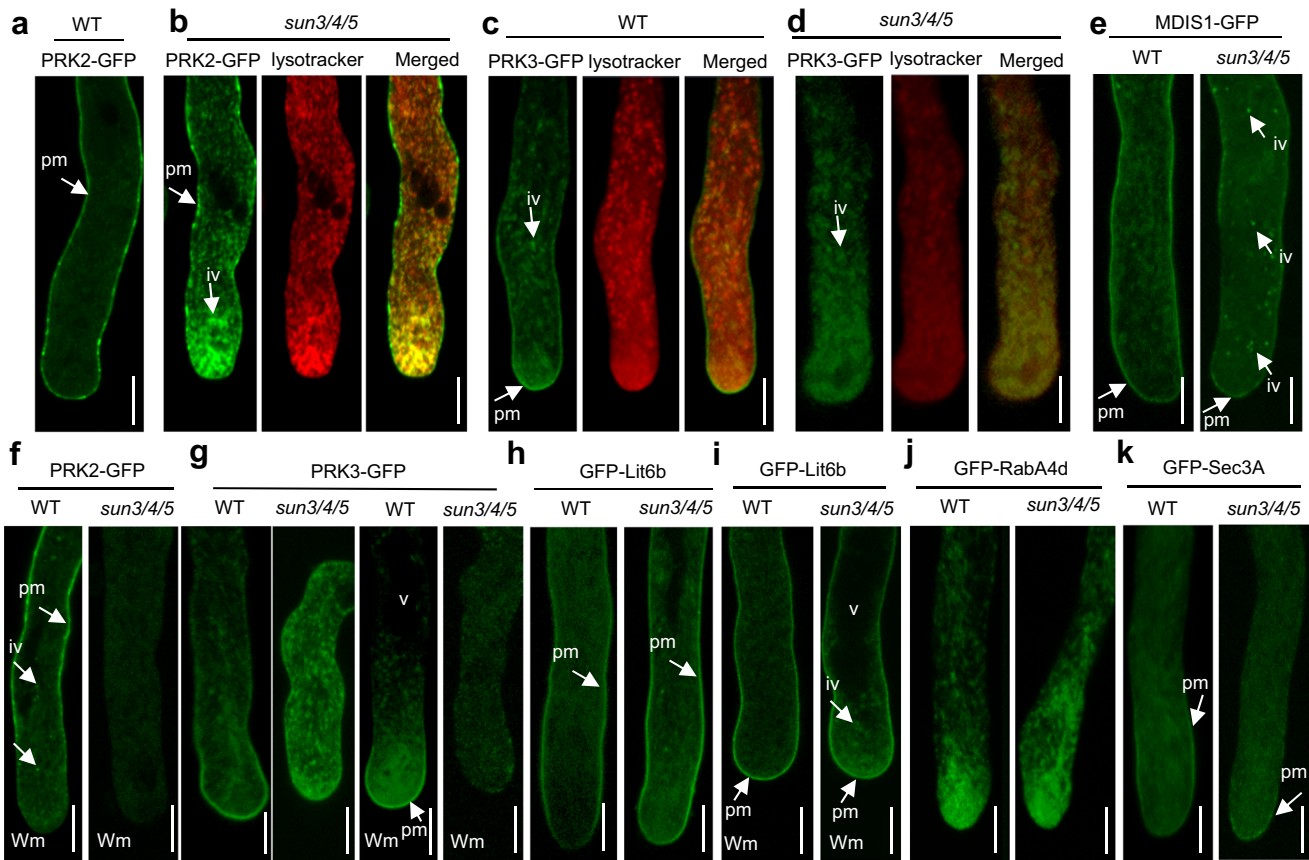

**Fig. 5 Localization of cargoes in *sun3/4/5* pollen tubes. a** Plasma membrane (pm) localization of PRK2-GFP in the WT. **b** PRK2-GFP enriched intracellular vesicles (iv) were stained with LysoTracker red in *sun3/4/5*. **c** PRK3-GFP is localized on the PM and intracellular vesicles in the WT. **d** The PM localization of PRK3 is impaired in *sun3/4/5*, and mainly localized in acidic vesicles stained by LysoTracker red. **e** The PM localization of MDIS1 was partially impaired in *sun3/4/5*. **f** Wortmannin (Wm) treatment disrupts the PM localization of PRK2-GFP at the apex and subapex region of WT pollen tubes. PRK2-GFP signal disappeared after Wm treatment in *sun3/4/5*. **g** In *sun3/4/5*, PRK3-GFP signal disappeared after Wm treatment, but in WT, the signal on the PM and vesicles remained. **h** The PM localization of GFP-Lit6b in *sun3/4/5*. **i** The PM localization of GFP-Lit6b after Wm treatment in *sun3/4/5*. **j** The vesicle localization of RabA4d at the apex region is not affected in *sun3/4/5*. **k** The PM localization of GFP-Sec3A is not affected in *sun3/4/5*. 3 (**a-k**) independent biological experiments were repeated. Bar, 5 μm. v, vacuole.

transcriptionally upregulated, suggesting activated unfolded protein response (UPR) in *sun3/4/5* (Supplementary Fig. 7). These results imply important function of SUN3/4/5 in ER folding or sorting of BRI1 and possibly other LRR-RKs.

LRR-RKs is a large gene family that are involved in perceiving various extracellular signals in plants[30]. Several LRR-RKs have been reported to be involved in pollen tube growth and targeting, such as PRK2, PRK3, and MDIS1[31,32]. To investigate the possible role of SUN3/4/5 in sorting of LRR-RKs in pollen tubes, the subcellular localization of PRK2, PRK3 and MDIS1 were examined, with Lit6b, a PM transmembrane protein without an extracellular domain, as a control[33,34]. PM localization of these proteins was partially abolished in *sun3/4/5* pollen tubes, and the fusion proteins were mainly localized in vesicular structures stained by LysoTracker red, like that of BRI1 (Fig. 5a–e and Supplementary Fig. 8A–D). In contrast to PRK2 and PRK3, the intracellular MDIS1-GFP-positive vesicles are fewer and tiny, although its PM localization was also partially abolished in *sun3/4/5* (Fig. 5e and Supplementary Fig. 8C). To examine the nature of the acidic vesicles, the pollen tubes were treated with Wortmannin (Wm), an inhibitor of PI3K that blocks protein sorting and recycling from prevacuolar compartment (PVC), as well as exo/endocytosis in pollen tubes[35]. In WT pollen tubes expressing PRK2-GFP, treatment with Wm for 20 min induced the formation of tiny intracellular punctate GFP signal and

decreased the GFP signal on the PM at the apex and subapical regions, which recapitulates the effect of Wm on exocytosis and endocytosis[35] (Fig. 5f). In *sun3/4/5*, the intracellular signal and the residual GFP signal on PM disappeared after Wm treatment, suggesting that the formation of the PRK2-enriched acidic vesicles in *sun3/4/5* depends on the prevacuolar trafficking (Fig. 5f). Similarly, for PRK3, Wm treatment reduced GFP signal on the PM of shank region in the WT pollen tube, while in *sun3/4/5*, the intracellular vesicles disappeared and the total intracellular GFP signal was reduced (Fig. 5g). These results imply that PRKs in *sun3/4/5* pollen tubes are sorted to the PVC route through the Golgi or ER-derived acidic vesicles, instead of the TGN-to-PM route.

Meanwhile, Lit6b, as well as RabA4d and Sec3A, components for vesicle trafficking, were also examined in the wild-type and *sun3/4/5* mutant. The PM localization of Lit6b was not affected in *sun3/4/5* and Wm treatment did not eliminate the fluorescence from the intracellular vesicles and plasma membrane (Fig. 5h, i). The endosome compartments labeled by RabA4d required for pollen tube growth exhibit no obvious decrease or pattern changes (Fig. 5j). Sec3A, a component of the exocyst that mediates membrane fusion between secretory vesicles and the PM, was also properly located on the plasma membrane at the pollen tube apex (Fig. 5k). Additionally, the secretion of PRK2-GFP to the PM was also impaired in half of the pollen tubes from

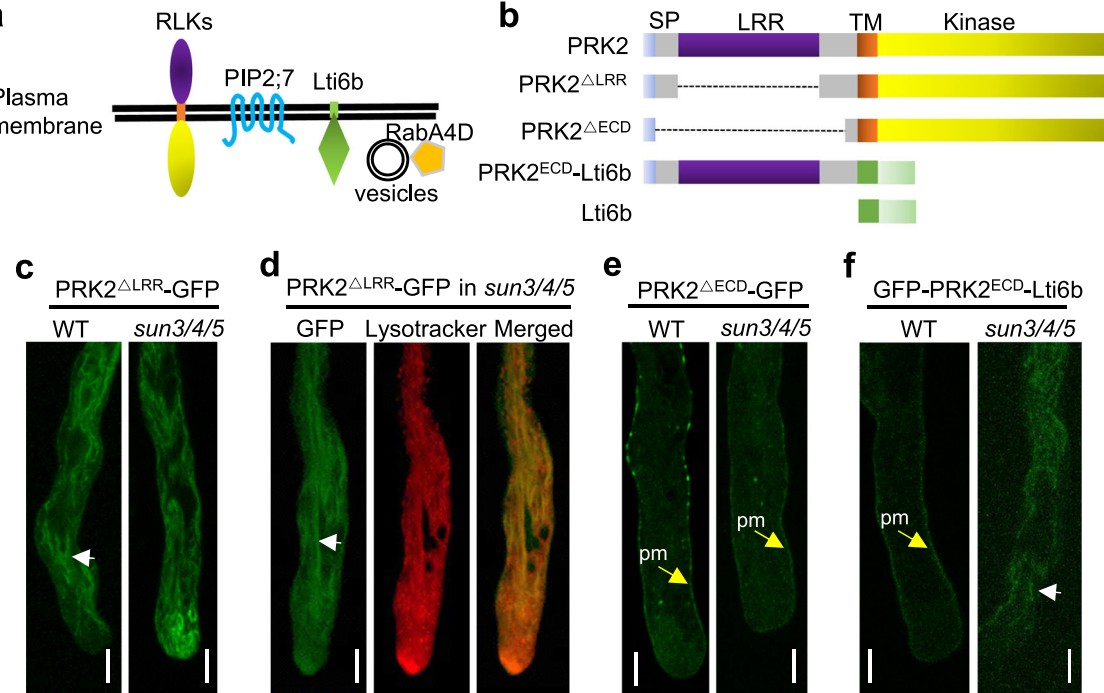

**Fig. 6 The proper sorting of PRK2 depends on the LRR domain. a** The topology of RKs, aquaporin, and Lti6b on the PM. **b** A diagram of the chimeric and deletion constructs of PRK2. **c** PRK2ΔLRR-GFP was mis-sorted to intracellular membrane structure in the WT and sun3/4/5. **d** PRK2ΔLRR-GFP was mis-sorted to acidic compartments stained by LysoTracker red. **e** PRK2ΔECD-GFP was sorted to the PM in WT and sun3/4/5. **f** GFP-PRK2ECD-Lti6b was missorted to membrane structures. 6 independent biological experiments were independently repeated for **c**–**f**. Bar, 5 μm. white arrows, membrane structures; yellow arrows, plasma membrane.

*pod1/+* plants ($n = 81$ pollen tubes), nevertheless, the intracellular GFP signal was feeble (Supplementary Fig. 8E), indicating protein degradation. Transmission electron microscopy (TEM) results showed normal Golgi and vacuolar structures, although the ER sheets appear thinner and longer than the wild-type (Supplementary Fig. 9). Although we failed to obtain the transgenic *sun3/4/5* plants expressing PRK6[32], the essential receptor for pollen tube attractant AtLURE1 in the parallel transgenic experiment, pollen tube attraction assay with AtLURE1.2[36] showed that the pollen tube attraction efficiency of *sun3/4/5* is reduced compared with the wild-type (Supplementary Fig. 10). In summary, these data suggest most likely that the secretion of LRR-RKs expressed in pollen tubes are all partially interfered, nevertheless the ER-Golgi/TGN-PM bulk secretion route still works in *sun3/4/5*. After ER exit, LRR-RKs is mainly mis-sorted to PVCs in *sun3/4/5*, although some are still sorted to the PM to sustain reduced plant growth and reproduction. According to the activated UPR response in *sun3/4/5*, mis-sorting of LRR-RKs should be caused by improper cargo processing in the ER and these cargos escape the ER-QC system without the guard of SUN3/4/5 but fail the surveillance of the QC systems in the following route, i. e. possibly in the Golgi, and are sorted to the PVC/vacuole for degradation.

To further investigate the cargo specificity of LRR-RKs governed by SUN3/4/5, the localization of LRR-RKs, PIP2;7 and Lit6b were examined in detail. Different from LRR-RKs that contain large extracellular LRR domains that face the ER lumen and undergo protein folding, PIP2;7 contains very few amino acids facing the ER lumen and Lit6b has no extracellular domain (Fig. 6a). This possibly implies that the ER luminal domains of the cargoes, which need ER folding and quality control, are involved in the SUN3/4/5-mediated sorting. To confirm this hypothesis, truncations and chimeric proteins by domain swapping were generated (Fig. 6b). Deletion of the LRR domain

of PRK2 caused the protein to be mis-sorted to the vacuole-like membrane structures, instead of the PM, in both WT and the *sun3/4/5* mutant (Fig. 6c). These membrane structures are acidic compartments as stained by Lysotracker red (Fig. 6d). This mis-sorting is possibly caused by the folding failure of the truncated extracellular fragment and is distinct from the cytosolic ERAD pathway for misfolded proteins. Surprisingly, when the whole extracellular domain (ECD) was deleted, leaving only the signal peptide (SP) and 20 amino acids preceding the TM, the protein was sorted to the PM in both the WT and *sun3/4/5* (Fig. 6e and Supplementary Fig. 8F). This indicates that the lack of the whole ECD let the protein free from the folding cycle and QC surveillance into the bulk flow secretion. When the TM and intracellular domain of PRK2 was replaced with that of Lti6b, the chimeric protein was properly sorted to the PM in the WT, but to the intracellular structures in *sun3/4/5* pollen tubes (Fig. 6f). These results suggest that the presence and proper folding of the LRR domain are involved in the SUN3/4/5-mediated protein sorting. For PRK2ΔLRR-GFP truncation and GFP-PRK2ECD-Lti6b chimeric protein, they are sorted into large tubular acidic compartments, but the full-length proteins are sorted to small acidic compartments. This different extent of mis-sorting suggests the possible involvement of the TM and kinase domain in combination with the LRR domain in cargo folding and sorting.

In summary, our data revealed an ER quality control mechanism for LRR-RK membrane receptors mediated by POD1-SUN3/4/5-CRT3 complex that is generally required for plant growth and reproduction.

## Discussion

In this study, SUN3, SUN4, and SUN5 were identified as POD1-associated proteins and play redundant roles in development. The physical interaction between POD1, SUN3/4/5, and CRT3, as well

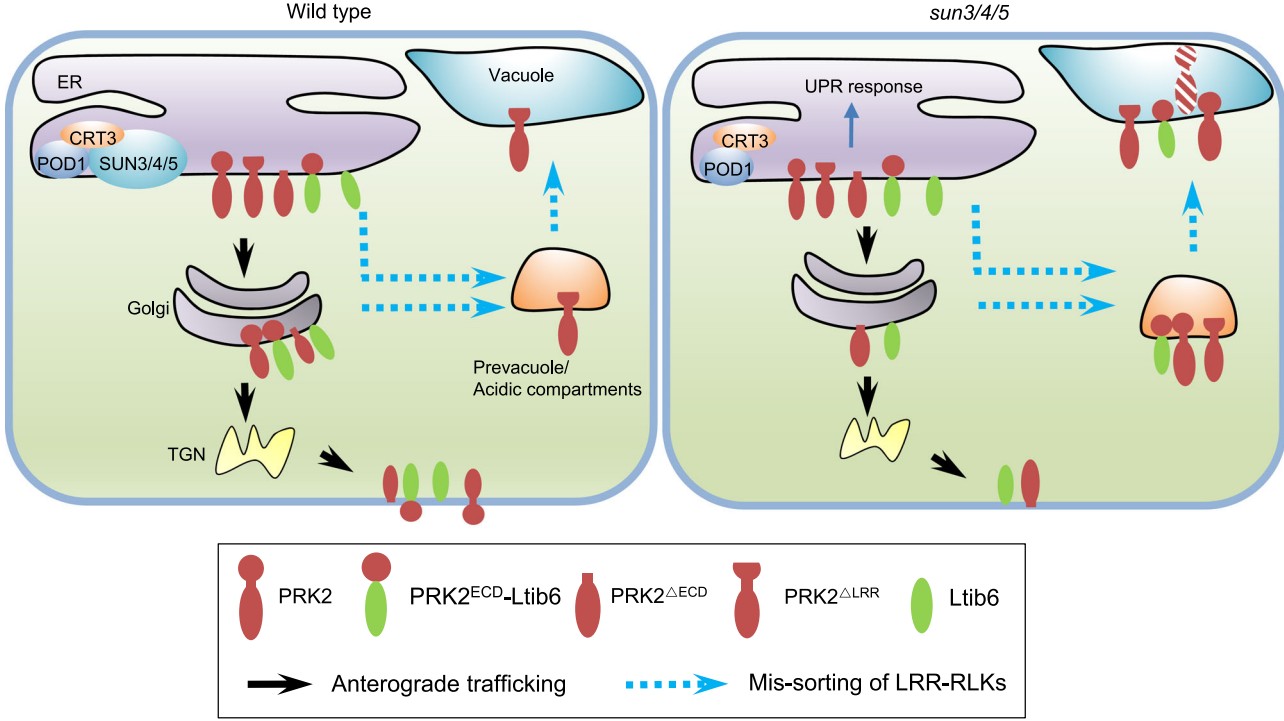

**Fig. 7 A working model of SUN3/4/5 in guarding the ER sorting of LRR-RKs.** SUN3/4/5-POD1-CRT3 on the ER membrane guards the quality control of LRR-RKs to ensure proper sorting of LRR-RKs and chimeric PRK2ECD-Ltib6 to the plasma membrane. Deletion of the LRR domain (PRK2ΔLRR) directs the truncated protein to the acidic compartments for degradation in the WT and *sun3/4/5*. In *sun3/4/5*, the full-length LRR-RKs and chimeric protein PRK2ECD-Ltib6 are missorted to the acidic compartments, like the effect of deletion of the LRR domain (PRK2ΔLRR). For LRR-RKs with all the extracellular domain deleted (PRK2ΔECD), they could be sorted to the plasma membrane.

as their phenotypic similarities suggest that they may be in the same complex in the early secretion of LRR-RKs, although the molecular activity of POD1 and SUNs, and how the three proteins are coordinated, are still unclear. The pleiotropic phenotype of *sun3/4/5* suggest that the function of SUN3/4/5 is general in the secretion of the large family of LRR-RKs that play critical roles in extracellular signal perception in plants. Several lines of evidence suggest that SUN3/4/5 play an important and specific role in the sorting of these cargoes. First, representative LRR-RKs required for vegetative and reproductive growth are mis-sorted from the ER to acidic compartments in roots and pollen tubes of *sun3/4/5*; Second, the PM localization of PIP2;7, and Lit6b was unaffected in *sun3/4/5*, indicating cargo specificity; Third, the affected cargoes did not undergo extensive ERAD, which extracts the misfolded proteins into the cytosol for proteasome-mediated degradation[37]; Fourth, LRR domain-deleted PRK2 and chimeric protein containing the extracellular domain of PRK2 are missorted in *sun3/4/5*. Furthermore, lack of the whole extracellular domain of PRK2 directs the protein to PM in *sun3/4/5*. These results suggest that SUN3/4/5-POD1-CRT3 (PSC) complex forms a guarding machinery at the ER membrane to promote the proper folding and sorting of LRR-RKs for anterograde trafficking. Without this complex, the plant cells deliver these cargoes to the acidic compartments for degradation (Fig. 7). This study establishes a checkpoint mechanism on the ER membrane for the secretory sorting of cell surface receptors.

Secretory proteins are scrutinized at the ER-to-Golgi station for their native or non-native state by the ER-QC. ER-localized chaperones monitor protein's folding state and prevent the aggregation of misfolded proteins which would be cleared by the ERAD pathway. In the absence of ER luminal chaperone CRT3, the defective BRI1 is released into the secretory pathway en route to PM and can properly function in ligand perception[14]. But in the

absence of SUN3/4/5, the LRR-RKs are released from the ER, but targeted to the acidic PVC/vacuole compartment for degradation. This suggests that the PSC complex acts in some checkpoints in the process of or following cargo folding mediated by the CRT/CNX QC system that can retain misfolded proteins in the ER. This also implicates the involvement of a post-ER quality control mechanism, likely the Golgi QC[37]. The post-ER quality control is important for cellular homeostasis, and its presence in plants is supported by the vacuolar mis-sorting of a structurally compromised PM-anchored glucanase[38,39]. Given these observations, two possible functions of SUN3/4/5 that are not mutually incompatible are raised. First, they could function in the checkpoint of the protein folding process, which is suggested by the final targeting to the acidic compartments, possibly due to the post-ER quality control. Second, they could function directly in the cargo sorting at the ER. Given the tight coupling of cargo folding and sorting, the PSC complex may function at the junction of these two processes.

The exact mis-sorting route of LRR-RKs to the acidic compartments in *sun3/4/5* still needs further investigation. In animal cells, misfolded proteins released from the ER transiently access the cell surface before targeting to the lysosome for degradation or direct release to vesicles, which fuse with the lysosome[40,41]. It is still unclear whether the RKs trafficked to the acidic compartments directly from the ER or from the Golgi. The RK-enriched acidic vesicles in *sun3/4/5* appear sensitive to Wortmannin that disrupts the post-Golgi vesicle trafficking, indicating a convergence at PVCs. In plants, multiple vacuolar sorting routes have been implicated, including Golgi-dependent and Golgi-independent, which are still quite uncharacterized[42,43]. In the microscopic observation of the fluorescent fusion proteins in *sun3/4/5*, the fluorescence of all missorted markers was weaker than that in the WT. This likely suggests that the mis-sorted cargoes undergo degradation in the acidic compartments, which is distinct from the cytosolic ERAD process.

The LRR domains of LRR-RKs determine the cargo specificity in SUN3/4/5-mediated protein targeting. Different cargoes undergo specific regulation of ER exit, which is possibly determined by cargo-specific amino acids, ternary structure, cargo size, topology, or other factors. PHF1, the structural analog of SEC12 that functions as the initiator of COPII vesicles, specifically regulates the ER export of phosphate transporter in plants[44]. In yeast, ER membrane protein p24 is involved in the sorting of GPI-anchored proteins by recruiting COPII components[45]. The effect of sun3/4/5 is distinct from the Arabidopsis loss-of-function mutant of p24, in which the GPI-anchored protein AGP4 is retained in the ER[46]. The SUN3/4/5-POD1 complex is present in yeast, animals, and plants, but the role is distinct, and the involvement of plant-specific CRT3 may contribute to the kingdom specificity. In yeast, the orthologous complex Slp1-Emp65 is specifically required to protect the folding intermediates of soluble proteins, but not membrane protein, from ERAD-mediated degradation, through direct binding to the polypeptides by the SUN domain[17]. In mouse, the ortholog OSPT is required for generating extracellular collagen, implying a possible role in secretion of soluble cargoes[18]. The evolutionary co-option of this complex in plants is likely driven by the gene expansion of LRR receptor kinases during the evolution of higher plants, accompanied by the emergence of land plant-specific CRT3. In summary, this study evidenced a mechanism that guards the ER exit of LRR-RKs. Future studies on how the PSC complex guards and monitors the folding and sorting of LRR-RKs will deepen our understanding of ER sorting mechanisms.

## Methods

**Plant materials**. T-DNA insertion lines, sun3/salk_093820, sun4-1/salk_022028, sun4-2/CS803680 and sun5/salk_126070 of Arabidopsis thaliana Col-0 ecotype, were obtained from the Arabidopsis Biological Resource Center.

**Constructs**. All plasmids were constructed by restricted enzyme-based ligation, Gateway recombination system, and Gibson assembly method. For the complementation constructs, SUN4 genomic fragments containing the promoter region (2.8 kb) and the transcribed region (including introns and exons), fused GFP, and SUN4 3'UTR fragments were inserted into pCAMBIA1300 vector, to generate gSUN4-GFP-TerSUN4. For pollen tube-specific expression of SUN4, CDS was inserted into pLAT52:GFP-TerNOS, to generate pLat52:SUN4-GFP-TerNOS. For BRI1-GFP, the genomic DNA fragment of BRI1 including 2725-bp promoter and 3591-bp CDS replaces the pLAT52 fragment in pCAMBIA1300-pLAT52:GFP-TerNOS, to generate pCAMBIA1300-pBRI1:BRI1-GFP-TerNOS. For 35 S promoter constructs, SUN3 and SUN4 CDS were inserted in to pCAMBIA2300-p35S:cGFP-TerOCS, generating pCAMBIA2300-p35S:SUN-GFP-TerOCS. For GFP reporter assay, the coding sequence of PRK2 was fused to GFP sequence and cloned into pCAMBIA1300, and LAT52 promoter and terminator of POD1 were assembled to generate pCAMBIA1300-LAT52:PRK2-GFP-TerPOD1. In the same way, Lit6b, Sec3A and RabA4d were inserted to the pCAMBIA1300-LAT52:GFP-TerPOD1. The coding sequences of PRK3 and MDIS1 driven by their native promoters were generated as pCAMBIA1300-PRK3:PRK3-GFP-TerPOD1 and pCAMBIA1300-MDIS1:MDIS1-GFP-TerPOD1. For truncated PRK2, the coding sequences were fused with GFP under the LAT52 promoter. All the plasmids were transformed into Agrobacterium tumefaciens GV3101 and transformed into Col-0 and the sun3/4/5 mutants. All the primers used in this study were listed in Supplementary Table 2.

**Phenotypic analysis**. For root phenotyping, 7-day-old WT and sun3/4/5 seedlings were stained in propidium iodide (10 µg/mL, Sigma) solution for 2 min in the dark, and then mounted on a slide and observed under a Zeiss confocal microscope (510 META) with 20× objective lens. Pollen grains were germinated on newly prepared germination media [1 mM CaCl$_2$, 1 mM Ca (NO$_3$)$_2$, 1 mM MgSO$_4$, 1 mM H$_3$BO$_3$, 18% (W/V) sucrose, 1% agarose]. For aniline blue staining, the pollinated pistils were fixed in ethanol/acetic acid (3:1) for 8–12 h, and then washed two times with PBS buffer. Then, 1 N NaOH was added and left for 12 h. After washing with PBS buffer twice, the samples were stained in 0.1% aniline blue for 12 h before microscopic observation with a Zeiss microscopy. For FM4-64 staining, a droplet of 5 µM FM4-64 (Invitrogen) was applied onto the germinated pollen tubes on the solid germination media for 5 min and washed out with liquid pollen germination media. The stained pollen tubes were observed with a Zeiss confocal microscopy with 543 nm excitation and 550–600 nm emission. For Lysotracker Red staining of root, 5-day-old seedlings were applied with Lysotracker Red (10 µg/mL in DMSO, Invitrogen) for 5–10 min

and washed out with liquid pollen germination media before confocal imaging with 543 nm excitation and 550–600 nm emission. Control treatment was performed with the same concentration of DMSO. For wortmannin (Wm) treatment, Wm (20 µM in DMSO, Santa Cruz Biotechnology) was applied to the root or pollen tubes for 20–30 min and washed out with liquid MS media liquid or pollen germination media, respectively. Pollen tube attraction assay with LURE1.2 was performed according to the reported[31,47]. The purified LURE1.2 peptides from bacteria E. coli was embedded in gelatin beads and applied to pollen tubes germinated in semi-in vitro condition. And then take photos every 30 min to record the growth direction.

**Confocal and electron microscopy**. Tobacco leaves infiltrated with indicated constructs were cut into ~9 mm$^2$ pieces, and then mounted on slides with sterilized water. Images were captured on a Zeiss LSM780 microscope equipped with argon-ion laser with 63× or 100× oil immersion objective lenses. The excitation/emission wavelength for GFP is 488 nm/510–530 nm. .

For electron microscopy, WT and sun3/4/5 roots and cotyledon of 5-day seedlings were excised and immersed in 50 mM Na-cacodylate buffer (pH 7.4), containing 1% paraformaldehyde, 2.5% glutaraldehyde, and 0.5% tannic acid, and then vacuum infiltrated for 0.5 h and kept at 4 °C overnight. The tissues were washed 5 times with Na-cacodylate buffer (pH 7.4), and post-fixated in 1% osmic acid for 3 h. Then, after washing 5 times with sterile water, the samples were dehydrated subsequently through an acetone series (30, 50, 70, and 90%) and twice in 100% acetone for 15 min each time at room temperature. Then after infiltrating stepwise in Epon812, the samples were embedded in molds. Polymerization was performed at 30 °C, 45 °C, and 60 °C for 24 h for each temperature, respectively. Ultrathin sections of 70 nm were cut on a Leica EM UC6 ultramicrotome (Leica) and collected on formvar-coated copper mesh grids. Sections were post-stained in uranyl acetate for 10 min and lead citrate for 5 min at room temperature. The grids were observed with electron transmission microscopy (JEM-1400) operating at 80 kV and micrographs were acquired with a system equipped with a CCD camera (Gatan823).

**Co-immunoprecipitation and immunoblotting**. For tunicamycin (TM) treatment, 5 µM TM (5 mM stock solution in DMSO) was used. After culturing for 16 h, the transformed protoplasts were lysed in extracting buffer (40 mM PIPES pH 7.4, 8 mM EGTA, 1 mM MgCl$_2$, 0.33 M sucrose, 1 mM DTT, 1 mg/mL cocktail protein inhibitors, 1 mg/mL PMSF, 1% Triton X-100) and kept on ice for 30 min. Meanwhile, the anti-Myc (Sigma) agarose or anti-GFP agarose (Chromotek) was pre-treated with extracting buffer. Then, the protoplasts lysis was centrifuged at 10,000 g for 10 min at 4 °C. The supernatant was incubated with anti-Myc agarose or anti-GFP agarose for 4 h at 4 °C. Then the agarose was washed with extracting buffer supplemented with 1% Triton X-100 for 6 times. Each sample was added to 50 µL 2 × SDS-PAGE loading buffer and boiled for SDS-PAGE electrophoresis. Immunoblots were performed with anti-GFP (1:2000 dilution, CWBIO, Cat. CW0086) and anti-Myc antibody (CWBIO, Cat. CW0299M) and second antibody, goat anti mouse (1:10,000, CWBIO, Cat. CW0102S).

**PNGase F treatment**. PNGase F (New England Biolabs) was used for the N-glycosylation assay according to instructions from the manufacturer. Briefly, 1 µL glycoprotein denature buffer (0.5% SDS and 40 mM DTT) was added to 1–20 µg total protein [quantified using the Bradford assay (Bio-Rad)] isolated from the plants and the volume was adjusted to 10 µL. Then, the sample was heated at 100 °C for 10 min to denature the proteins, and then put on ice for 1 min and subjected to centrifugation for 10 s. 2 µL glycobuffer (50 mM sodium phosphate, pH 7.5), 2 µL 10%NP40 and 6 µL water was added to the sample to adjust the volume to 19 µL. 1 µL PNGase F was added and mixed. Then the sample was incubated at 37 °C for 1 h and then mixed with the loading buffer for SDS-PAGE.

**Bioinformatics**. The protein sequences were retrieved from NCBI (https://www.ncbi.nlm.nih.gov/), TAIR (https://www.arabidopsis.org) and Phytozome (https://phytozome-next.jgi.doe.gov/). The protein domains were predicted on SMART (http://smart.embl-heidelberg.de/) and the transmembrane domain was predicted on TMHMM server (https://services.healthtech.dtu.dk/service.php?TMHMM-2.0). The phylogenetic tree was performed with MEGA X (https://www.megasoftware.net) with ML method and 1000 bootstrap. The protein names and sequences were included in Supplementary Data 1.

**Reporting summary**. Further information on research design is available in the Nature Research Reporting Summary linked to this article.

## Data availability

All data supporting the findings of this study are available within the manuscript and its supplementary files. Source data are provided with this paper.

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

## Acknowledgements

We thank the public technology service center of Institute of Genetic and Developmental Biology for assistance in confocal and electron microscopy, Dr. Yan Zhang (Shandong Agriculture University) for PRK2 plasmid, Dr. François Chaumont (Université Catholique de Louvain) for Venus-PIP2;7 plasmid, Dr. Jianmin Li for bri1-5 and bri1-9 seeds (University of Michigan). This work was supported by the National Natural Science Foundation of China (31991203 to W.Y. and 31270351, 31622010, 31870295 to H. L.).

## Author contributions

Conceived and designed the experiments: W.Y., H.L., Y.X., and J.M. Performed the experiments: Y.X., J.M., P.J., and Z.Z.; Analyzed the data and wrote the paper: H.L., Y.X., W.Y., and J.M.

## Competing interests

The authors declare no competing interests.
