## [Peer Review File · Nature Communications]

POD1-SUN-CRT3 chaperone complex guards the ER sorting of LRR receptor kinases in ArabidopsisREVIEWER COMMENTS

Reviewer #1 (Remarks to the Author):

This is an interesting study in the field of plant reproduction and the protein secretory pathway. Collectively, the authors have revealed a novel function of the SUN proteins in ER quality control through interacting with POD1, a key factor in protein folding. Different from SUN1/2 isoforms, SUN3/4/5 localized preferably to the ER network. The seed set rate in sun3/4/5 triple mutant was significantly reduced, a phenotype that directly related to the dysfunctional of male gametophyte. The reasons were not clear, but it is likely due to the mis-localization of multiple LLR receptors, which are essential in pollen signal transduction and guidance. On the other hand, the subcellular localization of BRI1 was also affected in sun triple mutant, results in dwarfed plants.

My immediate feeling of this study is, a lot of data, but lacks of data interpretation and explanation. The authors have found a specialised ER-QC and secretory pathway, and pleiotropic plant developmental phenotype, but the links between them is vague. Here are some major issues need to be addressed:

1. The Material and method section is too simple, a lot of key information is missing, which makes it very difficult to interpretate some of the results. For example, little information is given for the transgenic plants used in Figure 4 and Figure 5; without this information, some of the results and conclusion make little sense. Similarly, no information is given for PNGase F treatment, phylogenetic analysis, pollen tube staining ect. I would be happy to re-evaluate a re-submission once this information is provided.

Also, the labelling of this manuscript is awful. No line information is given in the main text, no labelling can be found on each figure. Clearly, a lot of efforts have been put into this work, it is a pity that the authors did not treat this submission very seriously.

2. What happens to SUN localisation at the endogenous expression condition? I would be interested to see if the ER localisation of SUN4 remains at the endogenous expression level. In figure 2F, the author used a SUN4-GFP driven by Lat52 promoter, where it localised in gSUN4-GFP/sun3/4/5 transgenic plants (also in vegetative tissue)?

It is known that over-expressing some NE protein also produce ER labelling. It is important to eliminate the possibility that the sun triple mutant phenotype is not because of the dysfunctional of nucleus. Could the author discuss this potential possibility and give a convincing explanation?

3. In figure 5 & 6, no information on lysotracker staining is given in M&M. I don't know which dye the authors have used here, at what concentration? Some of the these dye also labels mitochondria at high concentration. Please provide further information and careful controls.

4. The manuscript contains two separate studies, in plant reproductive and vegetative organ, but I did not find a strong link between them. I believe the authors tried to make the point that PRK proteins (which is the downstream receptor of LURE) mis-localisation in sun triple mutant give the pollen guidance phenotype and reduced seed set, especially at the basal part. However, key experiments are missing: what happens to PRK6 localisation? As PRK6 is more important in controlling pollen tube growth and LURE sensing than PRK2 or PRK3. Will sun mutant pollens response to LURE signal less effective? On the other hand, the author found the localisation of BRI1 is affected in sun triple mutant, and gives smaller plants. Could this phenotype be partially restored by addition BR signal to the growth medium?

Again, because the authors didn't explain their result enough, the take-home-messages for these studies are not clear.

5. The study on autophagy is too preliminary in Supplemental Fig. 7. There are several well-recognised methods to study autophagy flux rather than only showing the localisation of ATG8.

Besides, it does not make sense by transforming ATG9 into WT or sun3/4/5 to verify the autophagy activity. I believe this is not the focus of the current study, so please remove Figure S7 and related text. It doesn't add to this work.

6. The ER and vacuole labelling in Figure 4H, are not very clear, please provide better images.

7. Please quantify the signal intensity in Figure 5. What happens to pollen tube morphology and growth rate? A image on pollen in vitro germination assay will be helpful.

Minor issues:

1. Figure 1H, more background information is required, why protein glycosylation assays are performed, the intention is not clear.
2. Figure 2C, a label missing
3. What is POD1 and NRT3? How these proteins are related to the current study? More information is required in the introduction.
4. The authors stated that the sun3/4/5 mutants had short stamen filaments in Fig. 2A. As far as I can see, this phenotype may be caused by the extension of pistil, please explain.
5. It is mentioned that PIP2;7 is used as a control. Thus, the localization of PIP2;7 should be observed in the same region (DZ) of root cells in WT and sun3/4/5 mutants (Fig. 4G).

Reviewer #2 (Remarks to the Author):

The article by Xue et al describes the identification of SUN proteins as putative regulators of LRR protein trafficking to the plasma membrane. The authors show protein-protein interactions between POD1, a protein the previously identified, and SUN3,4 and 5 as well as CRT3. They propose that these proteins form a complex in the ER that mediates quality control for LRR receptors as they exit the ER towards the PM.

The authors present strong genetic data in support of a role for SUN proteins in pollen tube growth and fertilization. They also show synergistic effects of the sun and bri1 mutations indicating that SUN proteins may indeed regulate BRI1 traffic. They also show that SUN4 proteins localize to the ER. The weakest part of the paper is the interpretation of the role of SUN3/4/5 in LRR protein traffic as described below:

1) Results from Fig 5A to D could be interpreted differently than the authors viewpoint. First, PRK2-GFP and MDIS1-GFP still accumulate in the PM and perhaps at similar levels as the WT. In the absence of any quantification of fluorescent signals, one cannot conclude that the PM accumulation is decreased. In fig 5E for example, PM signal from MDIS1-GFP is not abolished as the authors indicate. There is clear enrichment at the tip and the upper half part of the image. These inconsistencies clearly warrant quantification of PM vs cytosolic signals. Second, the increased GFP fluorescence in the cytosol in Fig 5B, C and D could result from reduced degradation of proteins when they are targeted to the vacuole (after endocytosis). I don't think that the authors have sufficient evidence to separate the possibility of reduced PM traffic or enhanced vacuolar degradation.

2) The WM treatment is also problematic. WM treatment in the sun mutants results in very low GFP fluorescence (compare 5F with 5B), so it may be that is toxic in these cells. The interpretation that WM affected the formation of the acidic vesicles is not consistent with the almost complete loss of the PRK2-GFP protein (where did it go?). To demonstrate that these are meaningful experiments, it should be necessary to show that other cellular structures and proteins remain intact, for example, acidic organelles stained with lysotracker red and GFP-Lit1b. If these cells are hypersensitive to WM, it would be difficult to conclude its effect on a specific pathway.

3) The result from Fig 6E also needs to be quantified to show that there is significant PM accumulation in the sun mutants.

4) The authors nicely show interactions between protein pairs of POD1, SUN and CRT3. But the authors have not provided evidence that the phenotypes of the sun mutants on LRR protein trafficking are due to the activity of those 3 proteins in a complex. There are too many instances throughout the paper where this is already assumed, but clearly it is an over interpretation. SUN proteins could function in LRR protein traffic in the absence of POD1 and/or CRT3 interactions.

5) Fig 1A was already published in Graumann et al (slightly different version but same conclusions) and should be moved to Supplemental.

6) None of the published fluorescent markers (e.g. Lit6) have a source or a reference.

Minor issues:

-Fig 2A: The size of the two flowers shown is very different. In fact, the length of the stamens are similar, but the gynoecium is shorter in the WT. Please include images of flowers of the same size that better represent the mutant phenotype.

-Please include a reference for this statement: "rhodamine B hexyl ester which labels the ER in plant cells"

-The last line of page 8 should read WM instead of Tm.

-The resolution of Fig 6C is too low.

-Line 7 of page 13 PHT1 should read PHF1.

REVIEWER COMMENTS

Reviewer #1 (Remarks to the Author):

This is an interesting study in the field of plant reproduction and the protein secretory pathway. Collectively, the authors have revealed a novel function of the SUN proteins in ER quality control through interacting with POD1, a key factor in protein folding. Different from SUN1/2 isoforms, SUN3/4/5 localized preferably to the ER network. The seed set rate in sun3/4/5 triple mutant was significantly reduced, a phenotype that directly related to the dysfunctional of male gametophyte. The reasons were not clear, but it is likely due to the mis-localization of multiple LRR receptors, which are essential in pollen signal transduction and guidance. On the other hand, the subcellular localization of BRI1 was also affected in sun triple mutant, results in dwarfed plants.

My immediate feeling of this study is, a lot of data, but lacks of data interpretation and explanation. The authors have found a specialised ER-QC and secretory pathway, and pleiotropic plant developmental phenotype, but the links between them is vague. Here are some major issues need to be addressed:

1. The Material and method section is too simple, a lot of key information is missing, which makes it very difficult to interpretate some of the results. For example, little information is given for the transgenic plants used in Figure 4 and Figure 5; without this information, some of the results and conclusion make little sense. Similarly, no information is given for PNGase F treatment, phylogenetic analysis, pollen tube staining ect. I would be happy to re-evaluate a re-submission once this information is provided.

Response: The method section was revised as suggested. All the mentioned experiments, assay and plasmid constructs were described in detail.

Also, the labelling of this manuscript is awful. No line information is given in the main text, no labelling can be found on each figure. Clearly, a lot of efforts have been put into this work, it is a pity that the authors did not treat this submission very seriously.

Response: Labeling has been added in the revised figures and main text.

2. What happens to SUN localisation at the endogenous expression condition? I would be interested to see if the ER localisation of SUN4 remains at the endogenous expression level. In figure 2F, the author used a SUN4-GFP driven by Lat52 promoter, where it localised in gSUN4-GFP/sun3/4/5 transgenic plants (also in vegetative tissue)? It is known that over-expressing some NE protein also produce ER labelling. It is important to eliminate the possibility that the sun triple mutant phenotype is not because of the dysfunctional of nucleus. Could the author discuss this potential possibility and give a convincing explanation?

Response: SUN3, SUN4, SUN5-GFP driven by the endogenous promoters shows extremely low expression and cannot be observed with confocal microscopy in all

tissues examined, for example, roots, leaves, pollen and pollen tubes, although these constructs can rescue the mutant phenotype. This indicates that protein level of SUNs is under stringent control. That's why we expressed the fusion proteins driven by *35S* and *LAT52* promoter to examine their subcellular localization. Based on the absence of SUN4-GFP (driven by *LAT52* promoter) on the vegetative nucleus, SUN4 is localized on the ER even at this overexpression condition. We included this discussion in the revised manuscript.

3. In figure 5 & 6, no information on lysotracker staining is given in M&M. I don't know which dye the authors have used here, at what concentration? Some of these dye also labels mitochondria at high concentration. Please provide further information and careful controls.

Response: The details of the treatment were included in the revised Method. The concentration refers to published literatures on vesicle trafficking in plant cells.

4. The manuscript contains two separate studies, in plant reproductive and vegetative organ, but I did not find a strong link between them. I believe the authors tried to make the point that PRK proteins (which is the downstream receptor of LURE) mis-localisation in sun triple mutant give the pollen guidance phenotype and reduced seed set, especially at the basal part. However, key experiments are missing: what happens to PRK6 localisation? As PRK6 is more important in controlling pollen tube growth and LURE sensing than PRK2 or PRK3. Will sun mutant pollens response to LURE signal less effective? On the other hand, the author found the localisation of BRI1 is affected in sun triple mutant, and gives smaller plants. Could this phenotype be partially restored by addition BR signal to the growth medium?

Response: Transformation of *sun3/4/5* plants with PRK6-GFP was performed in parallel with all the other pollen tube markers. Unfortunately, the single transgenic plant showed no GFP signal in the pollen tubes. We believe that PRK6-GFP is also affected in *sun3/4/5* considering its homology to PRK2 and PRK3. In this manuscript, we have four LRR-RLK markers, BRI1, PRK2, PRK3, MDIS1, and two negative marker Lit1b and PIP2;7. And we also mapped the extracellular domain as the key to be required for this SUN-mediated sorting process. The major phenotype of *sun3/4/5* pollen tubes is slower growth rate, and pollen tube attraction defect was 25% higher than the wild type. Given these lines of evidence, PRK6-GFP was not paid special attention. Instead, in this revised version, we performed pollen tube attraction assay with LURE1.2 and found that *sun3/4/5* is less efficient in the response to this attractant than the wild type (Supplemental figure 10). This suggest that PRK6 should be also affected.

Addition of BR cannot rescue the phenotype of *sun3/4/5*, and *sun3/4/5* also shows reduced sensitivity to the high-concentration-induced hypocotyl elongation inhibition effect. This suggest reduced BR signaling in *sun3/4/5* quite likely caused by reduced PM presence of BRI1. This result was included in the revised Supplemental figure 6.

Again, because the authors didn't explain their result enough, the take-home-messages for these studies are not clear.

Response: Discussion of the integration of reduced PM presence of RLKs involved in both the vegetative and reproductive growth was included in the revised paragraph 1 in Discussion.

5. The study on autophagy is too preliminary in Supplemental Fig. 7. There are several well-recognised methods to study autophagy flux rather than only showing the localisation of ATG8. Besides, it does not make sense by transforming ATG9 into WT or sun3/4/5 to verify the autophagy activity. I believe this is not the focus of the current study, so please remove Figure S7 and related text. It doesn't add to this work.

Response: Figure S7 has been removed from the revised version.

6. The ER and vacuole labelling in Figure 4H, are not very clear, please provide better images.

Response: High-resolution images was provided in Figure 4H for the acidic compartments. The ER appears compact in root elongation zone due the smaller sized cells, so the ER pattern shown by ER staining is distinct from the leaf cells that exhibit clear polygonal ER networks in our hand and in literatures.

7. Please quantify the signal intensity in Figure 5. What happens to pollen tube morphology and growth rate? A image on pollen in vitro germination assay will be helpful.

Response: The image and statistics of pollen germination, tube growth were included in the revised Supplemental figure 3. Reduced pollen tube growth is the major phenotype of pollen tubes. And pollen tube morphology is normal as shown in all the confocal images of the marker lines, and pollen tube guidance defect on the micropyle is 25% (Supplemental figure 3).

Minor issues:

1. Figure 1H, more background information is required, why protein glycosylation assays are performed, the intention is not clear.

Response: N-linked protein glycosylation in the ER occurs to many ER cargos and resident proteins. PNGase F is an endoglycosidase that cleave the N-linked glycans from glycoprotein. The result showed that SUN4 is sensitive to PNGase F further indicate its ER localization and modification. This background and a reference were added to the revised main text.

2. Figure 2C, a label missing

Response: Revised accordingly.

3. What is POD1 and CRT3? How these proteins are related to the current study? More information is required in the introduction.

Response: POD1 is a conserved ER localized protein with unknown biochemical activity and was previously identified to be required for navigating pollen tubes to the ovules in Arabidopsis and interact with the ER chaperone calreticulin 3 (CRT3). More information on this background was included in the revised introduction.

4. The authors stated that the *sun3/4/5* mutants had short stamen filaments in Fig. 2A. As far as I can see, this phenotype may be caused by the extension of pistil, please explain.

Response: Usually, failed pollination due to shortened stamen filament lead to longer pistils after anthesis. So, in this revised Figure 2 and Supplemental figure 2, we provide new images and measured the filament length, ratio of filament/pistil, pistil length, with flowers right at the time of anthesis. The result show that the stamen filament of *sun3/4/5* is indeed longer than the wild type, but the pistil length is not affected.

5. It is mentioned that PIP2;7 is used as a control. Thus, the localization of PIP2;7 should be observed in the same region (DZ) of root cells in WT and *sun3/4/5* mutants (Fig. 4G).

Response: PIP2;7-Venus was under the *35S* promoter. However, we can not observe fluorescent signal on the division zone, whereas BRI1-GFP was mainly expressed and showed clearer signal.

Reviewer #2 (Remarks to the Author):

The article by Xue et al describes the identification of SUN proteins as putative regulators of LRR protein trafficking to the plasma membrane. The authors show protein-protein interactions between POD1, a protein the previously identified, and SUN3,4 and 5 as well as CRT3. They propose that these proteins form a complex in the ER that mediates quality control for LRR receptors as they exit the ER towards the PM.

The authors present strong genetic data in support of a role for SUN proteins in pollen tube growth and fertilization. The also show synergistic effects of the *sun* and *bri1* mutations indicating that SUN proteins may indeed regulate BRI1 traffic. They also show that SUN4 proteins localize to the ER. The weakest part of the paper is the interpretation of the role of SUN3/4/5 in LRR protein traffic as described below:

1) Results from Fig 5A to D could be interpreted differently than the authors viewpoint. First, PRK2-GFP and MDIS1-GFP still accumulate in the PM and perhaps at similar levels as the WT. In the absence of any quantification of fluorescent signals, one cannot conclude that the PM accumulation is decreased. In fig 5E for example, PM signal from MDIS1-GFP is not abolished as the authors indicate. There is clear enrichment at the tip and the upper half part of the image. These inconsistencies clearly warrant quantification of PM vs cytosolic signals.

Response: Plasma membrane targeting of LRR-RLKs is partially affected, as shown by BRI1, PRKs, and MDIS1. This indicates that the partial plasma membrane targeting in *sun3/4/5* can be executed by other SUNs or other pathways. The phenotype aggregation by combination of *sun3/4/5* and *crt3* supports this speculation. Totally abolishment of these LRR-RLK obviously would cause plant lethality, which may occur in *pod1/+*, which is embryo lethal. To verify this conclusion of reduced targeting, quantification

by fluorescence ratio of PM/cytosol of these images was included in the revised Supplemental figure 8.

Second, the increased GFP fluorescence in the cytosol in Fig 5B, C and D could result from reduced degradation of proteins when they are targeted to the vacuole (after endocytosis). I don't think that the authors have sufficient evidence to separate the possibility of reduced PM traffic or enhanced vacuolar degradation.

Response: All the confocal images for comparison between WT and *sun3/4/5* were obtained with the same setting. Fluorescence quantification suggest that the absolute PM signal is weaker in the mutant, and the PM/cytosol ratio of the signal is also reduced in the mutant (See above). In the wild type, BRI1-GFP undergoes very active constitutive recycling between the PM and endosomes to maintain an enrichment on the PM but not bulk flowed into the vacuole (Wang et al. Mol Plant, 2015; Rubbo et al. Plant Cell, 2013). It is quite possible that the increased cytosolic signal is due to reduced degradation in the vacuole. This consequence is caused by the defective PM targeting, which is one of the conclusions of the manuscript. Whether SUNs directly affect the vacuole degradation of these LRR-RLKs is still unclear, at least we have no evidence. Wortmannin inhibits exocytosis/endocytosis and trafficking from prevacuole back to TGN/Golgi. Wortmannin treatment suggest that PRK2 and PRK3 in intracellular vesicles in WT accumulate but degraded more rapidly in *sun3/4/5* than the WT. This result does not support defective degradation of PRKs in *sun3/4/5*.

2) The WM treatment is also problematic. WM treatment in the *sun* mutants results in very low GFP fluorescence (compare 5F with 5B), so it may be that is toxic in these cells. The interpretation that WM affected the formation of the acidic vesicles is not consistent with the almost complete loss of the PRK2-GFP protein (where did it go?). To demonstrate that these are meaningful experiments, it should be necessary to show that other cellular structures and proteins remain intact, for example, acidic organelles stained with lysotracker red and GFP-Lit1b. If these cells are hypersensitive to WM, it would be difficult to conclude its effect on a specific pathway.

Response: Parallel wortmannin experiments in Figure 5i showed that PM localization of GFP-Lit1b was not obviously affected, although the cytosolic signal appears increased, and the vacuole extends toward the pollen tube tip, which has been reported to be an effect caused by wortmannin in pollen tubes. The concentration of WM referred to published literatures that work on pollen tubes. Disappearance of PRK2-GFP and PRK3-GFP after WM treatment is a phenotype in *sun3/4/5* pollen tubes. Considering the known effect of WM in blocking of sorting and recycling, also shown in the wild type pollen tubes in Figure 5 f and g, the disappearance is quietly possibly due to protein degradation, possibly through the PVC/vacuole pathway. The references and discussion on this point were included in the revised version.

3) The result from Fig 6E also needs to be quantified to show that there is significant PM accumulation in the *sun* mutants.

Response: Quantifications was included in revised Supplemental figure 8.

4) The authors nicely show interactions between protein pairs of POD1, SUN and CRT3. But the authors have not provided evidence that the phenotypes of the sun mutants on LRR protein trafficking are due to the activity of those 3 proteins in a complex. There are too many instances throughout the paper where this is already assumed, but clearly it is an over interpretation. SUN proteins could function in LRR protein traffic in the absence of POD1 and/or CRT3 interactions.

Response: POD1 is quite possibly critical in the PM localization of LRR-RLKs, which is supported by the failed PM accumulation of PRK2-GFP in *pod1* pollen tubes as shown in Supplemental figure 8. Involvement of ER chaperone CRT3 in maturation and sorting of LRR-RLKs, such as BRI1 and EFR, has been reported, previously (Saijo Y, et al. 2009; Li, et al. 2009; Jin, et al. 2009). Our biochemical data showed that they may function together, but the molecular mechanism, and the exact biochemical activity of both POD1 and SUN3/4/5 is the future task. This was discussed in the revised discussion. In yeast, the two paralogs form a complex Slp1-EMP65 to guard the newly synthesized soluble protein, but not membrane proteins, in the ER from promiscuous ERAD, though direct binding of SUN domain to the unfolded client proteins (Zhang et al. Cell, 2017). However, the coordination mechanism between Slp1 and EMP65 is still unknown. Identification of this complex in yeast and plant in this study would open a new gate to dissect how the different arrays of client proteins are folded and sorted in the ER, which is still far from well understanding.

5) Fig 1A was already published in Graumann et al (slightly different version but same conclusions) and should be moved to Supplemental.

Response: Revised accordingly.

6) None of the published fluorescent markers (e.g. Lit6) have a source or a reference.

Response: Two references was included.

Minor issues:

-Fig 2A: The size of the two flowers shown is very different. In fact, the length of the stamens are similar, but the gynoecium is shorter in the WT. Please include images of flowers of the same size that better represent the mutant phenotype.

Response: See above.

-Please include a reference for this statement: “rhodamine B hexyl ester which labels the ER in plant cells”

Response: Revised accordingly.

-The last line of page 8 should read WM instead of Tm.

Response: Corrected accordingly.

-The resolution of Fig 6C is too low.

Response: The GFP signal in these vacuole-like membranes is relatively weak and highly dynamic with the pollen tube growth, so it quite hard to obtain high resolution

images. We have tried different treatment to slow down the movement, and dye staining to confirm the identity of these structures, while treatments usually disrupted the membrane integrity.

-Line 7 of page 13 PHT1 should read PHF1.

Response: Corrected accordingly.

REVIEWERS' COMMENTS

Reviewer #1 (Remarks to the Author):

The authors have addressed most of my concerns, but I am still not very convinced about the claimed ER localisation of SUN3/4/5 (Major 2). PNGase F treatment is not sufficient to exclude the NE possibility, as the outer nuclear envelope is continuous of the ER, it can produce protein, and have the same modifications.

In the past, K.Graumann JXB 2014 showed SUN3/4/5 interact with nuclear outer envelope proteins (KASH-domain proteins), and the mutants of sun exhibit defective in NE morphology, thus it is very likely that SUN3/4/5 is not solely ER. Given the pleiotropic phenotype of sun3/4/5 triple mutant in reproductive and vegetative tissue, I believe some of developmental defects in figure 3 & 4 are link to nuclear defective rather than protein mis-targeting. This possibility was conveniently ignored here.

As the authors not able to provide solid evidence (e.g. endogenous promoter driven expression or using an antibody), and I understand the technique difficulties here. Therefore, please discuss the possibility of the NE contribution here, and reflect better to the literature.

Reviewer #2 (Remarks to the Author):

The authors have addressed all my concerns. I was mostly concerned about the quantification of fluorescence intensities and those have been provided in SUPPL. Fig 8.

REVIEWERS' COMMENTS

Reviewer #1 (Remarks to the Author):

The authors have addressed most of my concerns, but I am still not very convinced about the claimed ER localisation of SUN3/4/5 (Major 2). PNGase F treatment is not sufficient to exclude the NE possibility, as the outer nuclear envelope is continuous with the ER, it can produce protein, and have the same modifications.

In the past, K.Graumann JXB 2014 showed SUN3/4/5 interact with nuclear outer envelope proteins (KASH-domain proteins), and the mutants of sun exhibit defective in NE morphology, thus it is very likely that SUN3/4/5 is not solely ER. Given the pleiotropic phenotype of sun3/4/5 triple mutant in reproductive and vegetative tissue, I believe some of developmental defects in figure 3 & 4 are linked to nuclear defects rather than protein mis-targeting. This possibility was conveniently ignored here.

As the authors are not able to provide solid evidence (e.g. endogenous promoter driven expression or using an antibody), and I understand the technical difficulties here. Therefore, please discuss the possibility of the NE contribution here, and refer better to the literature.

Response: A sentence of discussion on the possible function of SUN3/4/5' s on the nuclear envelope was added to the revised manuscript as "Thus, it cannot be excluded that SUN3/4/5 also function on the nuclear membrane, which is continuous with the peripheral ER" on page 6 (highlighted red).